# Stability of Carboxyl-Functionalized Carbon Nanotubes in Simulated Cement Pore Solution and Its Effect on the Compressive Strength and Porosity of Cement-Based Nanocomposites

Laura Silvestro [1],* , Geannina Terezinha Dos Santos Lima [2] , Artur Spat Ruviaro [3] and Philippe Jean Paul Gleize [3]

1    Department of Civil Engineering, Federal University of Technology-Paraná (UTFPR), Guarapuava 85053-525, PR, Brazil
2    Laboratory of Waste Valorization and Sustainable Materials (ValoRes), Department of Civil Engineering, Federal University of Santa Catarina (UFSC), Florianópolis 88040-900, SC, Brazil; geanninasantos@hotmail.com
3    Laboratory of Application of Nanotechnology in Civil Construction (LabNANOTEC), Department of Civil Engineering, Federal University of Santa Catarina (UFSC), Florianópolis 88040-900, SC, Brazil; arturspatruviaro@gmail.com (A.S.R.); p.gleize@ufsc.br (P.J.P.G.)
*    Correspondence: laurasilvestro@utfpr.edu.br

**Abstract:** The application of carbon nanotubes to produce cementitious composites has been extensively researched. However, the dispersion of this nanomaterial remains a technical limitation for its use. Thus, initially, this study assessed the stability of carboxyl-functionalized CNT on aqueous suspensions and simulated cement pore solution for 6 h through UV–visible spectroscopy. Subsequently, a CNT content of 0.1% by cement weight was incorporated into the cement pastes, and the compressive strength after 7, 14, 28, and 91 days was evaluated. In addition, the porosity of the CNT cementitious composites at 28 days of hydration was investigated by mercury intrusion porosimetry (MIP), and the microstructure was evaluated via scanning electron microscopy (SEM). The simulated cement pore solution's alkaline environment affects the CNT stability, progressively reducing the dispersed CNT concentration over time. CNT reduced the cementitious matrix pores < 50 nm by 8.5%; however, it resulted in an increase of 4.5% in pores > 50 nm. Thus, CNT incorporation did not significantly affect the compressive strength of cement pastes. SEM results also suggested a high porosity of CNT cementitious composites. The CNT agglomeration trend in an alkaline environment affected the CNT performance in cement-based nanocomposites.

**Keywords:** carbon nanotube; cement; pore solution; compressive strength; porosity

## 1. Introduction

Carbon nanotubes (CNT) have been extensively studied to improve the mechanical properties, durability, and specific applications of cement-based materials, such as monitoring structures, due to their intrinsic self-sensing properties [1,2]. Most studies focused on using nonfunctionalized CNT in cementitious matrices [3]. Nonetheless, noncovalent and covalent functionalization strategies can be adopted to improve the CNT dispersion, one of the significant problems in applying the nanomaterial.

The CNT covalent functionalization process can be divided into direct sidewall or defect functionalization. The direct covalent sidewall functionalization can be attributed to the modification of the hybridization from $sp^2$ to $sp^3$. On the other hand, defect functionalization is associated with the defect sites, such as open ends, holes in the CNT sidewalls, and irregularities in the hexagon graphene structure [4]. Ma et al. [4] mentioned that oxygenated sites can also be classified as defects. Thus, the oxidation process is usually

conducted with acids (e.g., sulfuric and nitric acids). In these cases, the defects are stabilized by hydroxyl (–OH) and carboxyl (–COOH) groups on the CNT surface [5].

The carbon of nonfunctionalized CNT walls is chemically stable, which results in an inert material and a low efficient load transfer across the CNT/matrix interface. Within this framework, the covalent functionalization can enhance the interfacial interaction with the matrix and, consequently, its properties [4]. Moreover, it can improve the CNT solubility and dispersion in solvents and polymers [6]. Furthermore, the covalent functionalization is particularly interesting since various chemical reactions can be carried out through the –COOH and –OH groups on the CNT surface [6].

Nevertheless, covalent functionalization can affect the CNT properties. For instance, the defect sites increase the scattering of electrons and phonons, which are associated with the electrical and thermal properties of these nanomaterials. Thus, CNT can tolerate a limit of defects such that significant changes in mechanical and electronic properties do not occur [7]. In addition, using concentrated acids and strong oxidants has negative impacts from an environmental point of view [4].

Previous studies are inconclusive regarding the effect of CNT functionalization with –COOH and –OH groups on the mechanical properties of cement-based materials. Some researchers reported increases in compressive and flexural strength [8,9], while others reported insignificant modifications or decreases in the mechanical properties compared to nonfunctionalized CNT [10,11].

Although CNTs functionalized with polar groups are more hydrophilic and, therefore, have a better dispersion in water, the dispersion in the cement matrix differs in the presence of a highly alkaline environment [12]. For instance, Zou et al. [13] observed different optimal ultrasonication values for CNT dispersion in water (determined by UV–visible spectroscopy) and in a cementitious matrix considering the mechanical properties. The polar groups (–COOH and –OH) may interact with the ions from the pore solution, delaying the cement hydration process and the strength development [14]. Additionally, the carboxyl groups of CNTs can make the environment more acidic, increasing the cement matrix's porosity [15]. Musso et al. [16] also mentioned that carboxyl-functionalized CNT has a hydrophilic behavior and can adsorb the mixing water, hindering cement hydration.

The study conducted by Mendoza et al. [17] investigated the effect of calcium hydroxide ($Ca(OH)_2$) on the stability of OH-functionalized CNT to simulate the Portland cement hydration environment. It was observed that $Ca(OH)_2$ interacted with the negative charge of OH–CNT, which negatively affected the electrostatic dispersion of CNT, resulting in an agglomeration trend. According to Li et al. [18], little attention has been given to this topic and the re-agglomeration of CNT with –COOH and –OH groups due to their interaction with $Ca^{2+}$ ions in the cement matrix. The influence of a simulated Portland cement hydration environment on the stability of carboxyl-functionalized CNT has not yet been thoroughly investigated. Thus, firstly, this study assessed the stability of COOH–CNT on the simulated cement pore solution using the UV–visible spectroscopy technique. Subsequently, the effect of the incorporation of 0.1 wt.% CNT on the compressive strength, porosity, and microstructure of cement pastes was investigated.

## 2. Materials and Methods

### 2.1. Materials

Carboxyl-functionalized multiwalled carbon nanotubes (CNTs) were used. The CNTs were supplied by Nanostructured and Amorphous Materials, and their properties are shown in Table 1. Figure 1 exhibits a scanning electron microscopy image of the CNTs obtained using a VEGA3 (TESCAN, Brno-Kohoutovice, Czech Republic) microscope operating at 15 kV, which shows the agglomeration trend of the nanomaterial. Ordinary Portland cement (OPC) was used for cement paste production. Moreover, a superplasticizer was also employed (MC-PowerFlow, Mc-Bauchemie). A complete characterization of all the materials used in this research can be obtained in previous studies published by the authors [12,19–21]. For the simulated cement pore solution, calcium sulfate dihy-

drate (≥99.0%, Neon, São Paulo, Brazil), potassium hydroxide (≥85.0%, Laffan, São Paulo, Brazil), sodium hydroxide (≥98.0%, Sigma-Aldrich, St. Louis, MO, USA), and calcium hydroxide (≥95.0%, Nuclear, São Paulo, Brazil) were used.

**Table 1.** Carboxyl-functionalized CNT properties.

| Inside Diameter (nm) | Outside Diameter (nm) | Length (μm) | SSA (m²/g) | Purity | –COOH Content (%) |
|---|---|---|---|---|---|
| 5–10 | 20–30 | 10–30 | >200 | 95% | 1.9–2.1% |

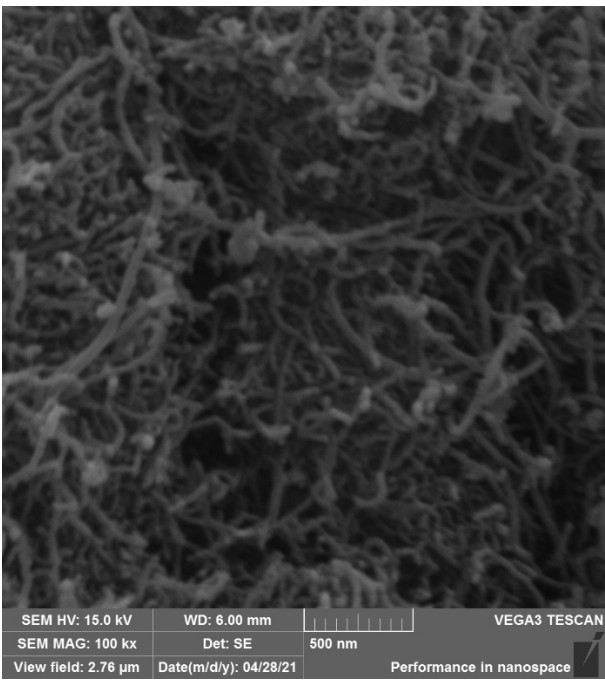

**Figure 1.** SEM image of CNTs at 100,000× magnification.

*2.2. Methods*

2.2.1. UV–Visible Spectroscopy

The stability of CNT dispersions in water and the simulated cement pore solution was evaluated through UV–visible spectroscopy. The correlation between absorbance and CNT concentration was obtained by applying a similar procedure to that described by Gao et al. [22] and Li et al. [23]. Initially, the CNT, deionized water, and superplasticizer were weighed, mixed, and sonicated in a tip sonicator (Vibra-Cell, VCX Serie, 750 W, 20 kHz, Sonics & Materials Inc., Newtown, CT, USA), with a diameter of 13 mm and an amplitude of 50% for 6 min [24,25]. Then, well-dispersed suspensions with 2.5 mg/mL CNT (sonicated as previously described) were diluted in deionized water at ratios of 1:50, 1:75, 1:100, 1:125, and 1:150. Thus, the extinction coefficient value corresponded to the slope of the linear correlation that passed through the zero intercept between the absorbance at the wavelength of 253 nm and the maximum theoretical concentration of the CNT suspensions ($C_t$). In aqueous solutions, the characteristic absorption of CNT appeared at 253 nm; accordingly, it was the absorbance value adopted for calculating the CNT concentration [26].

After that, CNT dispersions with a concentration of 2.5 mg/mL and a CNT:SP ratio of 1:2 [19] were prepared via sonication, as previously described. Subsequently, the compositions were centrifuged for 5 min (4000 rpm) to obtain a well-dispersed CNT solution [27]. Subsequently, the compositions were diluted in deionized water or simulated pore cement solution (composition further described) in the proportion of 1:100 with magnetic stirring for 5 min to keep the absorbance values in the range of 0.1–2 [28]. Dilution is necessary

because, in solutions with low concentrations of CNT, the absorbance obtained by the UV–visible spectroscopy can be linearly related to the CNT concentration, as described by the Lambert–Beer law. Thus, the dispersed CNT ($C_d$) concentration can be calculated using Equation (1). The absorbance values were determined until 6 h after dispersion via sonication, with measurements each hour to assess the carboxyl-functionalized CNT stability in water and the simulated pore cement solution.

$$C_d = \frac{A}{\varepsilon \times l}, \tag{1}$$

where A is the absorbance at 253 nm wavelength, $\varepsilon$ is the extinction coefficient, and l is the length of the optical path of light through CNT suspensions (l = 1 cm) [23].

The simulated pore solution adopted was based on the composition reported by Gao et al. [22]: 8 g/L sodium hydroxide (NaOH), 22.4 g/L potassium hydroxide (KOH), 27.6 g/L calcium sulfate dihydrate ($CaSO_4 \cdot 2H_2O$), and a saturated solution of $Ca(OH)_2$.

UV–visible spectroscopy analysis was carried out in a UV-5100S digital spectrophotometer. The absorbance measurements were obtained as a single scan, with 0.5 nm intervals, in the analysis range of 200 to 600 nm, using 10 mm quartz cuvettes. In the reference cuvette, deionized water was used [19].

### 2.2.2. Cement Pastes

CNT aqueous dispersions composed of 0.1 g of CNT, 40 g of water (equivalent to a CNT concentration of 2.5 mg/mL, as previously evaluated), and a CNT:SP proportion of 1:2 were prepared via sonication following the procedure described in Section 2.2.1. Subsequently, the CNT aqueous dispersions were added to 100.0 g of OPC and mixed in a high-shear mixer (10,000 rpm) for 3 min.

The axial compressive strength of cement pastes was evaluated at 7, 14, 28, and 91 days. Six cylindrical specimens (diameter of 19 mm and height of 26 mm) were molded for each age evaluated. The samples were kept in a cure submerged in water until the test date. Subsequently, the cylindrical specimens were tested in an Instron Model 5569. An analysis of variance (ANOVA) of the results was performed using Origin software.

The mercury intrusion porosimetry test was performed in an AutoPore IV (Micromeritics) equipment, which can determine the pore size distribution from 0.003 to 360 μm. The test was carried out in cement paste cubes measuring approximately $10 \times 10 \times 10$ mm, which were tested at 28 days. For the analysis, the surface tension of mercury of 0.485 N/m and a contact angle of 130° were considered.

The scanning electron microscopy (SEM) test was performed on cement paste samples after 28 days of hydration. The hydration reactions of the cement paste fragments were interrupted with the exchange of solvent using isopropanol. The samples remained immersed in isopropanol for 7 days and then dried in an oven at a temperature of 40 °C to avoid damage to the microstructure of the cement matrix. Images were acquired on a VEGA3 (TESCAN, Brno–Kohoutovice Czech Republic) microscope operating at 15 kV. For analysis, the samples were placed on carbon tape and covered with a thin layer of gold.

## 3. Results and Discussion

### 3.1. UV–Visible Spectroscopy

Figure 2 shows the calibration curve, which correlates the absorbance values determined by UV–visible spectroscopy at a wavelength of 253 nm with the CNT concentration. As previously described, $\varepsilon$ was determined by a linear fitting between absorbance and CNT concentration, considering a zero-intercept line. The regression equation exhibited in Figure 2 indicates that $\varepsilon = 37.1839$ mL·mg$^{-1}$·cm$^{-1}$, with good correlation ($R^2 = 0.996$). This value is in agreement with the coefficient $\varepsilon = 39.92$ mL·mg$^{-1}$·cm$^{-1}$ observed by Li et al. [23] for multiwalled carbon nanotubes dispersed in water. Moreover, the extinction coefficient determined in this study is close to the experimental data reported in previous research [22,29,30].

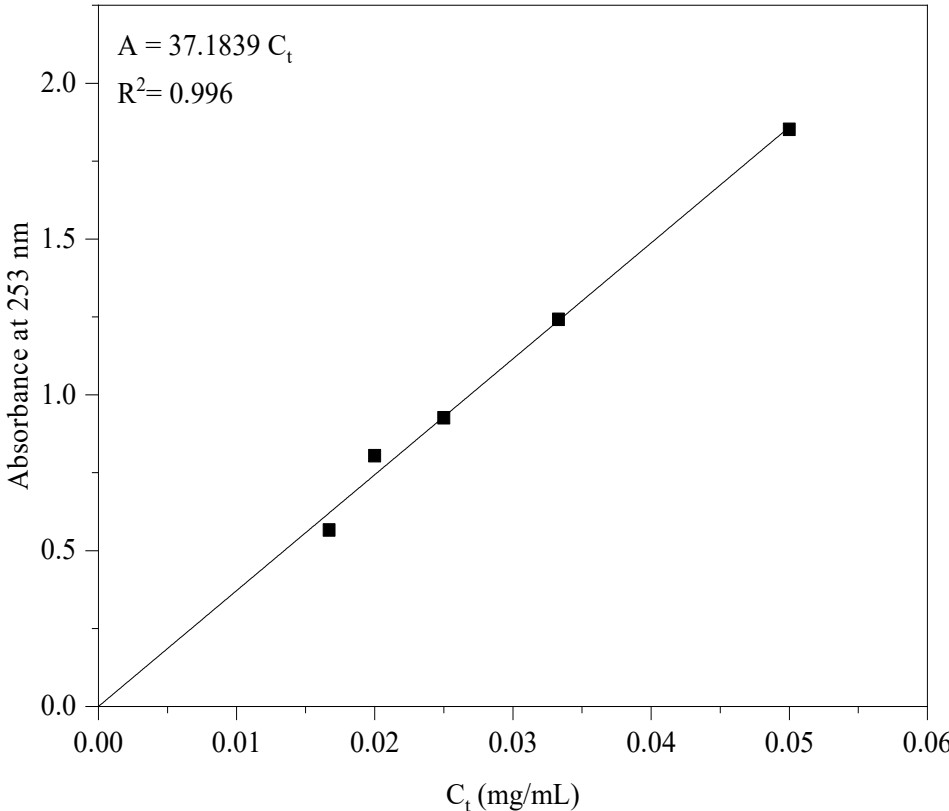

**Figure 2.** Correlation between absorbance at 253 nm and CNT concentration.

After the determination of the calibration curve, the stability of the CNT dispersion in water was evaluated for 6 h, as shown in Figure 3. During the 6 h analyzed, the CNT dispersed in water did not show a significant change in stability since the dispersed concentration concerning the initial total concentration remained unchanged. Previous research reported similar results. For instance, Collins et al. [31] observed that nonfunctionalized aqueous dispersions of CNT with a polycarboxylate-based superplasticizer presented good stability after 9 days. Garg et al. [32] also observed that a polycarboxylate superplasticizer enhanced the dispersion of aqueous carboxyl-functionalized CNT solutions. Sezer and Koç [33] assessed the stability of CNT treated with sulfuric and nitric acids in deionized water after sonication through photographs, reporting good stability even after 3 months.

Considering that the limiting factor for CNT application is precisely its dispersion, this result is interesting for the application of the material since aqueous dispersions of CNT can be produced in the laboratory for later application. Nonetheless, as previously discussed, the CNT dispersion in water does not represent its dispersion in a cementitious matrix. Thus, as also exhibited in Figure 3, the stability of aqueous dispersions of CNT was also evaluated in a simulated cement pore solution. In contrast to the behavior in water, there was an expressive reduction in dispersed CNT concentration over time. The $C_d/C_t$ reduced from 0.92 (0 h) to 0.44 (6 h). Note that, in the initial 30 min, the dispersions also presented a reduction in the dispersed CNT of 12.0%.

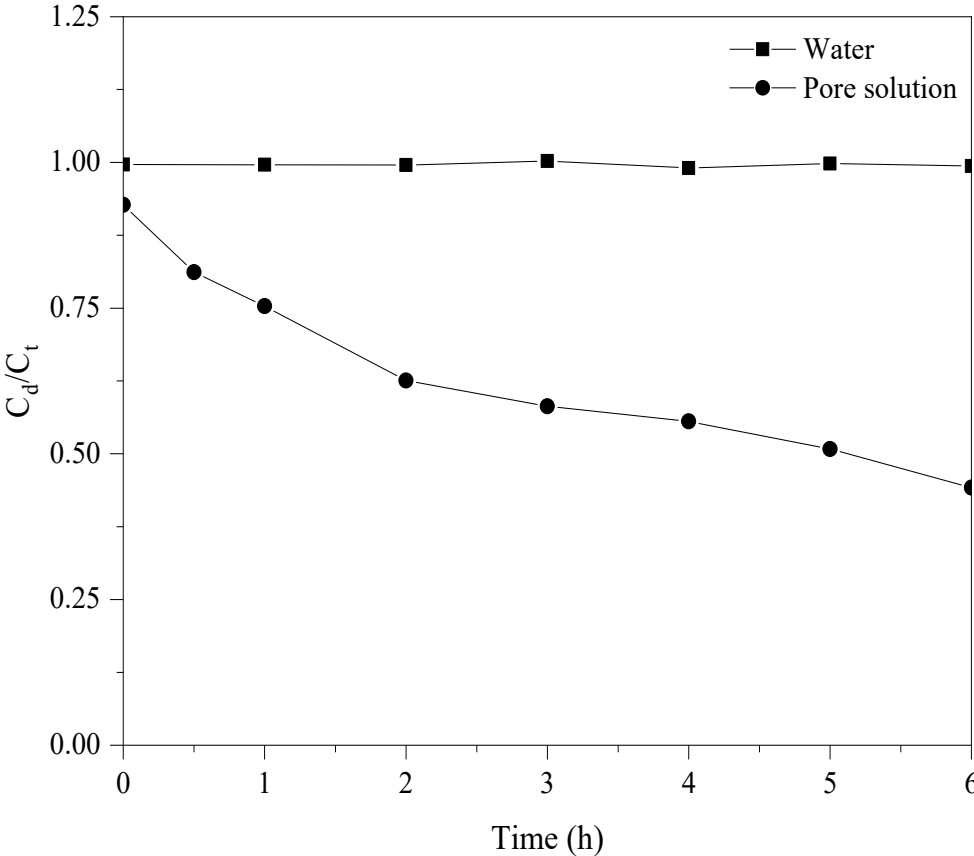

**Figure 3.** Stability of CNT dispersions in water and simulated cementitious pore solution after 6 h.

Similarly, Gao et al. [22] mainly observed the re-agglomeration and sedimentation of CNT after 3 h. In an alkaline pore solution, the $C_d/C_t$ of CNT progressively decreased, reaching values lower than 40% after 18 h of exposure. Li et al. [18] also observed poor dispersion stability, with stratification after 3 h of low-temperature plasma-modified CNT in a pore solution (pH 12.8) obtained by centrifugation from cement hydration products. The $Ca(OH)_2$ environment prevents the SP adsorption onto CNT surface, affecting its stability. In fact, as suggested by the FTIR spectrum, CNT's functional groups interact with $Ca(OH)_2$ [17]. According to Sabziparvar et al. [34], calcium ions show higher interaction energy with the oxygen groups than monovalent ions, such as sodium and potassium.

### 3.2. Compressive Strength

The compressive strength of plain cement paste (0.0% CNT) and CNT cementitious composite (0.1% CNT) after 7, 14, 28, and 91 days of hydration is shown in Figure 4. Regardless of the hydration time, no statistical differences in the compressive strength of cement pastes were observed with the incorporation of CNTs. If not adequately dispersed, the CNT agglomerates can impair the mechanical enhancement promoted by the nanomaterial addition. This trend corroborates the UV–visible spectroscopy results regarding the stability of aqueous CNT dispersions in simulated cement pore solution. The alkaline environment during cement hydration favors the agglomeration trend of CNT, helping to justify these compressive strength results. Moreover, the energy applied in the sonication process may be insufficient to properly disperse this CNT content. Similar results were obtained by Rhee et al. [35] regarding the addition of COOH–CNT to OPC mortars. For instance, mortar with a CNT content of 0.8 wt.% showed compressive strength values of 22.2, 27.0, and 33.1 MPa after 7, 14, and 28 days of hydration. In turn, plain mortar exhibited compressive strength values of 19.5 MPa (7 days), 22.9 MPa (14 days), and 31.8 MPa (28 days).

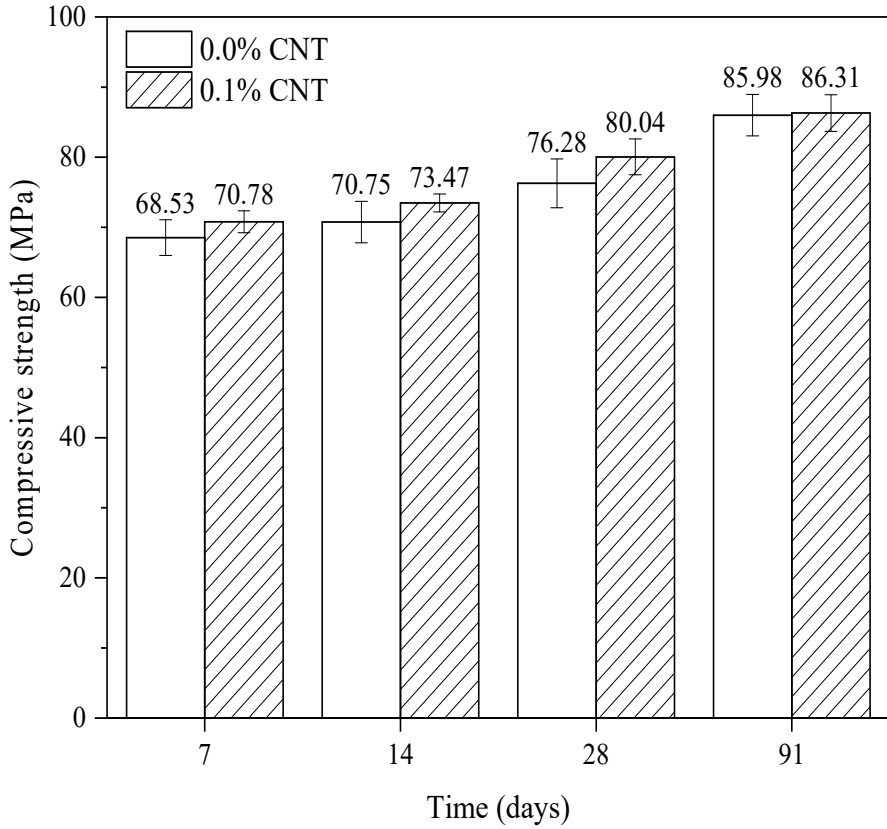

**Figure 4.** Compressive strength of cement pastes after 7, 14, 28, and 91 days.

### 3.3. Porosity

One of the most used techniques to characterize the pore structure of hydrated cementitious materials is mercury intrusion porosimetry (MIP). The basic principle of this technique is the intrusion of mercury into the porous structure by applying increasing pressure to the sample [36]. Figure 5 shows the cumulative intrusion volume and pore size distribution of cement pastes after 28 days of hydration. Table 2 summarizes the main results of the MIP test. Overall, cementitious matrix pores can be classified into mesopores (<50 nm) and macropores (>50 nm) [37]. The CNT incorporation slightly reduced the mesopore volume from 0.0350 to 0.0322 mL/g, while increasing the macropore intrusion volume from 0.0671 to 0.0701 mL/g. The macropores have a more significant effect on the mechanical and impermeability properties of cement pastes [38]. Thus, although CNT refined the mesopore structure of cement pastes, it increased the macropore volume. Hu et al. [38] also reported a more expressive influence of carboxyl-functionalized CNT on the mesopores of cement-based materials than in the macropores. In turn, Li et al. [39] observed reductions of 32.9% and 44.9% in the meso- and macro-porosity, respectively, upon incorporating COOH–CNT. Isfahani et al. [40] produced OPC mortars with a CNT content of 0.1 wt.%, which resulted in a porosity reduction of 12.8% compared to plain mortar (without CNT). According to Reales et al. [41], CNT modifies the porosity of CNT–cementitious composites due to a nucleation and filler effect. The authors also mentioned that this modification usually occurs on a mesopore scale. In this study, the total porosity of cement-based materials was not significantly affected by the CNT incorporation, which is in line with the compressive strength results previously discussed.

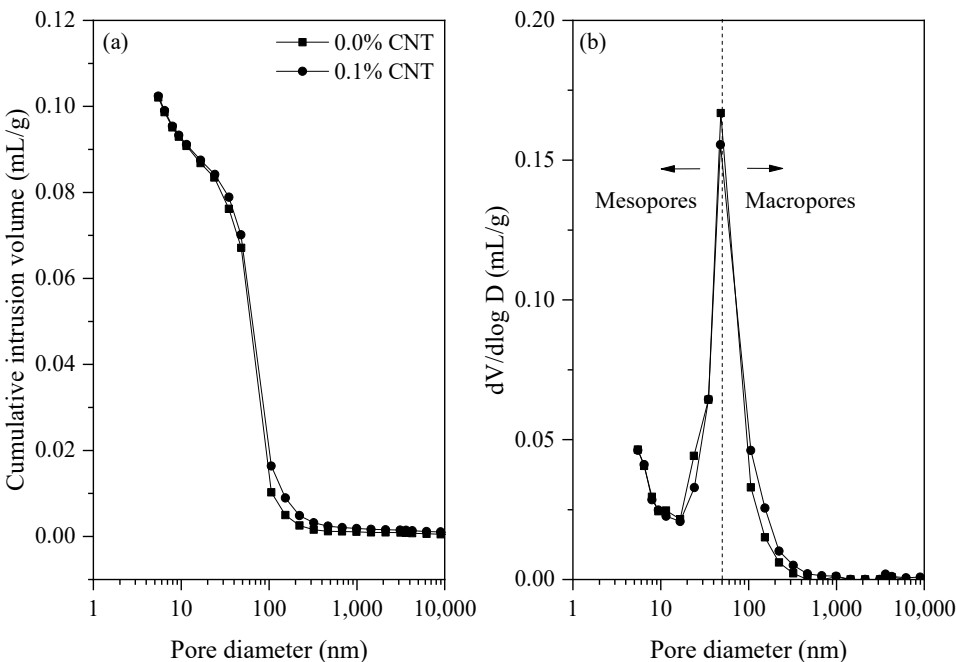

**Figure 5.** Cumulative pore volume (**a**) and pore size distribution (**b**) of cement pastes after 28 days.

**Table 2.** Cumulative intrusion volume, average pore diameter, and porosity of cement pastes after 28 days.

| Cement Paste | Cumulative Intrusion Volume (mL/g), d >50 nm | Cumulative Intrusion Volume (mL/g), d < 50 nm | Average Pore Diameter (nm) | Porosity (%) |
|---|---|---|---|---|
| 0.0% CNT | 0.0671 | 0.0350 | 31.6 | 15.11 |
| 0.1% CNT | 0.0701 | 0.0322 | 32.9 | 15.06 |

The microstructure of cement pastes was assessed by SEM, as shown in Figure 6. The images also suggest slightly higher macropores (indicated by the arrow) in the 0.1 wt.% CNT–cementitious composites compared to the plain cement paste, in line with the MIP results previously discussed.

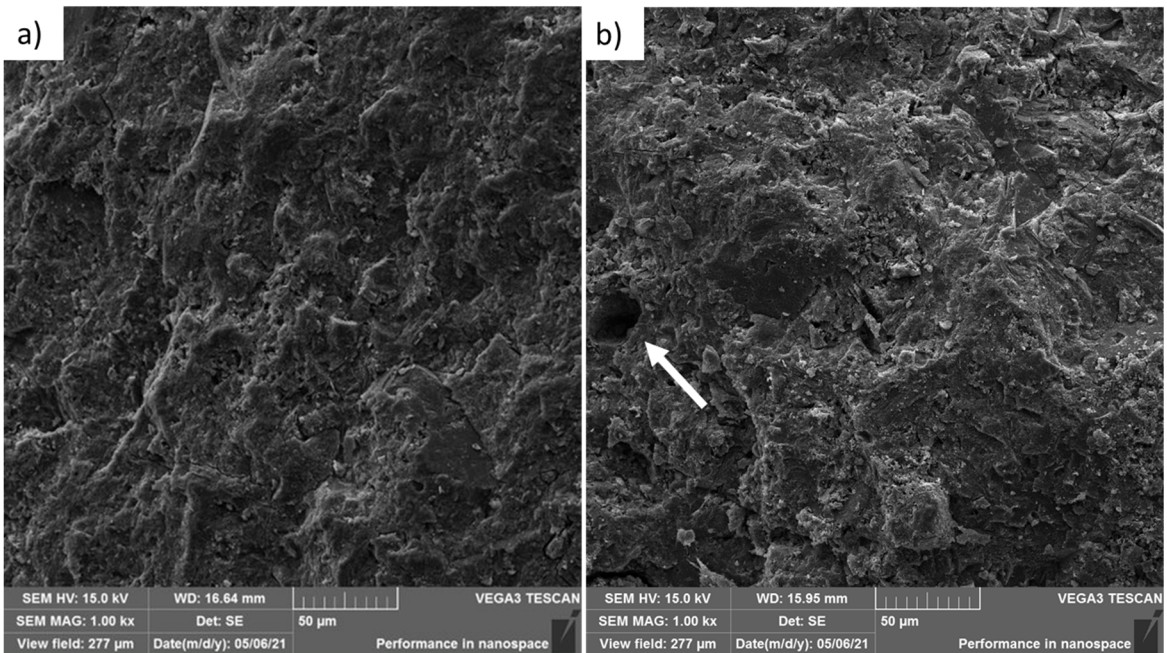

**Figure 6.** SEM images of cement pastes after 28 days: (**a**) 0.0% CNT and (**b**) 0.1% CNT (1000×) magnification).

## 4. Conclusions

This study assessed the stability of carboxyl-functionalized CNTs in water and simulated pore cement solution over 6 h. Moreover, it investigated the effect of 0.1 wt.% CNT on the compressive strength, porosity (MIP), and microstructure (SEM) of cement pastes. The carboxyl-CNT exhibited good stability in water, mainly due to the stabilization promoted by the polycarboxylate-based SP. However, the dispersion showed a significant loss of stability in the simulated cement pore solution. In future studies, the stability of aqueous CNT dispersions in a pore solution extracted by centrifugation of a cement paste should be evaluated. Additionally, it is recommended to evaluate CNT dispersions in cement pore solutions composed of supplementary cementitious materials to investigate how the composition of the cement matrix can affect the stability of nanomaterials. Concerning the compressive strength, no statistical differences were observed with CNT addition regardless the age. This was associated with a reduction in mesopores with CNT incorporation but with an increase in the intrusion volume of macropores. This study showed the importance of considering the effect of the cement pore solution on the stability of aqueous CNT dispersions, and it revealed that the dispersion in water is not representative of the dispersion in the cement matrix.

**Author Contributions:** Conceptualization, L.S.; methodology, L.S.; validation, L.S.; formal analysis, L.S., G.T.D.S.L. and A.S.R.; investigation, L.S., G.T.D.S.L. and A.S.R.; resources, P.J.P.G.; data curation, L.S.; writing—original draft preparation, L.S., G.T.D.S.L. and A.S.R.; writing—review and editing, L.S., G.T.D.S.L., A.S.R. and P.J.P.G.; visualization, L.S.; supervision, P.J.P.G.; project administration, P.J.P.G.; funding acquisition, P.J.P.G. All authors have read and agreed to the published version of the manuscript.

**Funding:** This research received no external funding.

**Institutional Review Board Statement:** Not applicable.

**Informed Consent Statement:** Not applicable.

**Data Availability Statement:** Not applicable.

**Acknowledgments:** The authors acknowledge the Brazilian governmental research agencies CNPq, CAPES, and FAPESC. Afonso Azevedo is acknowledged for the mercury intrusion porosimetry test. Patricia Prates and LABMAT-UFSC are acknowledged for the SEM images.

**Conflicts of Interest:** The authors declare no conflict of interest.

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
