# Peer review of "Stability of Carboxyl-Functionalized Carbon Nanotubes in Simulated Cement Pore Solution and Its Effect on the Compressive Strength and Porosity of Cement-Based Nanocomposites"

_carbon_

Round 1
Reviewer 1 Report
Present work is on the dispersion effect of functionalized cnt in cement pore solution and its effect to the properties of cement based nanocomposite.
It is an interesting work and could be suitable for the journal.
There are several points that need to be addressed.
i) Pore diameter and SEM measurements should be made for all the samples and compared between them.
ii) Please explain how the period after hydration was selected as 7, 14, 28, and 91 days.
iii) Please explain why compressive strength improves just by storing the cement pastes for many days.
iv) Photo of dispersed samples for each hour help understand the dispersion state.
v) Please explain why CNT addition had no effect on mechanical property of the cement composite.
Author Response
The authors are grateful for the Reviewers’ comments. All aspects listed were carefully revised and corrected. Our replies are presented below, immediately following your questions. Please note that changes from the previous version are seen in red in the revised manuscript. The number of lines cited in the answers is related to the revised manuscript.
i) Pore diameter and SEM measurements should be made for all the samples and compared between them.
Reply: We agree with the reviewer. A more in-depth analysis of the pore diameter and SEM images would be interesting, especially in all evaluated ages. However, it would demand the investigation of a large number of samples and the acquisition of many images. Unfortunately, this analysis will not be possible, especially considering that the Laboratory is a multi-user laboratory with typically long waiting lines for analysis. However, we emphasize that the influence of nanotubes on matrix porosity followed the same trend for all ages evaluated. Thus, the behavior verified for the samples at 28 days of hydration represents the other assessed ages (7, 14, and 91 days).
ii) Please explain how the period after hydration was selected as 7, 14, 28, and 91 days.
Reply: We appreciate the reviewer's inquiry. In fact, these ages are usual in the study of cementitious materials and are well established in the literature. For example, several standards set requirements based on a hydration age of 28 days.
Please explain why compressive strength improves just by storing the cement pastes for many days.
Reply: The hydration process of Portland cement occurs from its contact with water and develops continuously over time. The increase in the compressive strength of the cement pastes analyzed in this study is associated with this continuous hydration process. This behavior has already been widely reported in the literature, and it can be said that it is a consolidated knowledge about the material, so it was not discussed in the manuscript since the study's objective was to precisely evaluate the influence of CNT on the cement matrix. However, we appreciate the reviewer's questioning and hope to have answered your question.
iv) Photo of dispersed samples for each hour help understand the dispersion state.
Reply: We agreed with the reviewer that the images would be interesting for the study. However, when the test was performed, the images were not recorded. However, it is noteworthy that quantifying the concentration of dispersed NTC by UV-Vis spectroscopy is a more complete and reliable analysis than visual analysis.
v) Please explain why CNT addition had no effect on mechanical property of the cement composite.
Reply: As discussed in the manuscript, the dispersion of COOH-CNT in contact with ions during the cement hydration process has its stability affected. Thus, the tendency to agglomerate the CNTs compromised the reinforcing capacity and, thus, the increments in the mechanical properties of the cementitious matrices. This is discussed in lines 226-229 of the manuscript.
Reviewer 2 Report
This study which was performed in great detail has evaluated the stability of carboxyl-functionalized carbon nanotube in water and simulated pore cement solution over 6 hours as well as the effect of a CNT content of 0.1 wt.% on the compressive strength, porosity, and microstructure (by SEM) of cement pastes. The carboxyl-CNT exhibited good stability in water due to stabilization promoted by the polycarboxylates. However, the dispersion showed a significant loss of stability in simulated cement pore solution. In future the authors suggest studies of the stability of aqueous CNT dispersions in a pore solution extracted by centrifugation of a cement paste and also evaluate CNT dispersions in cement pore solutions composed of supplementary cementitious materials, to investigate how the composition of the cement matrix can affect stability. The paper is well written and presented.
Author Response
The authors are grateful for the Reviewers’ comments. All aspects listed were carefully revised and corrected. Our replies are presented below, immediately following your questions. Please note that changes from the previous version are seen in red in revised manuscript. The number of lines cited in the answers is related to the revised manuscript.
i) This study which was performed in great detail, has evaluated the stability of carboxyl-functionalized carbon nanotube in water and simulated pore cement solution over 6 hours as well as the effect of a CNT content of 0.1 wt.% on the compressive strength, porosity, and microstructure (by SEM) of cement pastes. The carboxyl-CNT exhibited good stability in water due to stabilization promoted by the polycarboxylates. However, the dispersion showed a significant loss of stability in simulated cement pore solution. In future the authors suggest studies of the stability of aqueous CNT dispersions in a pore solution extracted by centrifugation of a cement paste and also evaluate CNT dispersions in cement pore solutions composed of supplementary cementitious materials, to investigate how the composition of the cement matrix can affect stability. The paper is well written and presented.
Reply: We appreciate the review and reviewer comments.